# Understanding dietary behaviour change after a diagnosis of diabetes: A qualitative investigation of adults with type 2 diabetes

Roshan R. Rigby[1,2☯]*, Lauren T. Williams[1,2‡], Lana J. Mitchell[1,2‡], Lauren Ball[1,2‡], Kyra Hamilton[2,3☯]

1 School of Health Sciences and Social Work, Griffith University, Southport, Gold Coast, QLD, Australia,
2 Menzies Health Institute Queensland, Southport, Gold Coast, QLD, Australia, 3 School of Applied Psychology, Griffith University, Mt Gravatt, QLD, Australia

☯ These authors contributed equally to this work.
‡ LTW, LJM and LB also contributed equally to this work.
* roshan.rigby@griffithuni.edu.au

**Data Availability Statement:** All relevant data are within the manuscript and its Supporting Information files.

## Abstract

Type 2 diabetes (T2D) is a major public health concern. Optimal management of T2D often requires individuals to make substantial changes to their dietary intake. This research employed a qualitative methodology to examine decision making processes underpinning dietary behaviour change. Semi-structured telephone interviews were conducted on a purposive sample of 21 Australian adults who had recently consulted a dietitian after being diagnosed with T2D. Data were analysed using theoretical thematic analysis and themes were matched deductively with constructs that underpin motivational, volitional, and implicit processes which exist in common models of behaviour change. Influences on motivation, such as a *desire to improve health status* and *making use of valuable support networks* featured in participant narratives. Volitional influences included *knowing their limits*, *dealing with falling off the wagon*, and *learning how their body responds to food*. The themes *unlearning habits* and *limit the availability* were identified as underpinning implicit influences on dietary change. *Individual differences* and *emotions* were constructs additional to the model that influenced dietary change. These findings contribute to a richer understanding of the subjective experiences of adults with T2D regarding dietary change and highlight the multiple processes that guide their decision making in this context.

## Introduction

Non-communicable diseases, such as Type 2 Diabetes (T2D), substantially contribute to the global burden of morbidity and mortality [1] and are inextricably linked to 'lifestyle' behaviours such as poor diet [2]. A systematic review exploring the dietary intake of people with T2D found that most do not eat according to national recommendations for food groups [3]. Lifestyle intervention for the treatment and management of T2D recommends that health professionals provide education and prescription on dietary change as well as advice and knowledge on self-management strategies for disease control and health behaviours such as physical

**Funding:** This research did not receive any specific grant from funding agencies in the public, commercial, or non-for-profit sectors. This project was supported by the Australian Government Research Training Program. The funders had no role in study design, data collection and analysis, decision to publish, or preparation of the manuscript.

**Competing interests:** The authors have declared that no competing interests exist.

activity [4]. However, to effectively elicit positive outcomes, these lifestyle interventions require individuals to be committed and motivated to change [5].

It is well documented that individuals' beliefs and motivations affect their initiation and maintenance of health behaviours [5, 6], including dietary behaviours [7–9], and that providing education and advice alone is unlikely to be sufficient in achieving behaviour change [10]. Dietetic care is effective in diabetes management [11, 12], and primary health care settings present a crucial area to deliver lifestyle interventions for adults with T2D. The recommendations made by the International Diabetes Federation and the Australian National Evidence Based Guideline for Patient Education in Type 2 Diabetes focus on structured diabetes education that has multidisciplinary involvement [4, 13] and is patient-centred [14]. These guidelines present a key opportunity for health professionals to support patients in their behaviour change by addressing social psychological factors and motivational drivers of individual behaviour.

Dietary behaviour changes, and indeed health behaviours more generally, are complex [5]. Integrated models of behaviour change can be used in research to explore influences on health behaviour and ways to change them [9, 15]. Such models are usually underpinned by theories of motivation and social cognition to identify determinants of behaviour and examine the processes by which those determinants may relate to behaviour [16]. Hagger and Chatzisarantis proposed the integrated behaviour change (IBC) model to incorporate key constructs that underpin multiple processes known to guide behavioural decision making [17]. The model outlines the factors that relate to the motivational processes that lead to intention formation [18, 19], the self-regulatory and volitional processes that facilitate the translation of intentions to behaviour [20, 21], and the non-conscious, implicit processes that lead to behavioural engagement beyond an individual's awareness [22]. Using such integrated models, and adopting a methodology that reflects individuals' subjective experiences and perspectives about behavioural changes made after diagnosis, may provide a comprehensive picture of influences specific to dietary change in people with T2D.

Although integrated models of behaviour change show promise in providing a comprehensive description of behavioural determinants and the associated processes by which they act across various health behaviours [7–9], they do not account for all likely behavioural determinants. For example, the IBC does not take account of individual difference factors and emotion determinants, despite considerable research indicating the importance of these constructs to behavioural decision making [23, 24]. The exploration of such factors, alongside motivational, volitional, and implicit determinants of the IBC model, may pave the way for more elaborated integrated theories to be tested to advance knowledge on behavioural determinants and processes [16].

Nutrition research has used the IBC model in the context of fruit and vegetable consumption [7], adolescent sugar-sweetened beverage consumption [8], and predicting sugar consumption [9]. A recent systematic review exploring the use of behaviour change theories in dietetic practice found that there is little diversity in theories used within dietary interventions suggesting the need to explore other models in the dietetic context to align with the progressing health psychology literature [25]. T2D presents a case of complex behavioural constructs and, to date, the IBC model has not been used among adults to understand their dietary behaviours after a diagnosis of T2D, nor have other factors that underpin affective processes and individual difference factors been explored alongside propositions of the IBC model. Using a comprehensive integrated theoretical framework to guide study design and analysis is a new approach to understanding dietary behaviour change and is also important in the further development of integrated theories and assisting in improving model comprehensiveness.

One way of understanding the multiple processes guiding dietary behaviours is by exploring the subjective experiences of people (for an overview see Braun & Clarke, 2013 [35]). Qualitative methods can provide a rich understanding of underlying processes guiding individual behaviour and discover new ideas and unpack novel or poorly understood constructs that can be tested in further formative research. In the context of T2D, qualitative exploration can provide in-depth insight into how behaviour change processes may shape individuals' self-enacted changes and changes they enact as a result of dietary advice. Previous research on T2D and dietary advice have explored patient-centred care from the perspective of patients and dietitians. Findings from these studies highlight differences between patient and dietitian views on dietary change or shared decision making, in that dietitians might not always perceive patients to be the best advocate in their own dietary care which may impact treatment delivery and adherence [26, 27]. Other qualitative research has revealed people with T2D prefer individualised dietetic care that is not prescriptive [28]. Thus, uncovering individuals' perspectives on processes leading to dietary change after a diagnosis of T2D could lead to a better understanding of nutrition care utilising effective behaviour change methods and techniques. This study aimed to explore the decision-making processes of dietary behaviour change after a diagnosis of T2D in adults with T2D, focusing on the multiple motivational, volitional, implicit, and emotion and individual difference factors that may have influenced their decisions. Findings will provide an understanding of and formative evidence on key constructs that underpin dietary behaviour change post a diagnosis of T2D.

## Method

### Design

An interpretive paradigm guided the research using a relativist ontological view [29]. Being relativists, the researchers understand that reality is constructed intersubjectively and developed socially and experientially [29]. The relativist viewpoint was demonstrated by gaining the perspective of people about their individual dietetic consultations and the dietary changes they made, which have been given meaning based on their own experience and view on reality. The researchers have a subjectivist epistemological position, as knowledge is created subjective to those who experience it [29]. Knowledge was obtained by actively being involved in the research process [29]. An interpretive descriptive methodology was used [29]. Therefore, theoretical thematic analysis was conducted to give meaning to the data.

Semi-structured telephone interviews were conducted with free-living individuals (i.e., not part of a controlled intervention) across Australia to explore the experience of consulting with a dietitian after a new diagnosis of T2D. The data collection took place between October 2019 and February 2020 in Southeast Queensland, Australia, in a workplace setting with no other researchers or nonparticipants present. The Consolidated criteria for Reporting Qualitative research (COREQ) checklist and the APA Journal Article Reporting Standards for qualitative research [30] guided the reporting of this qualitative study (see S1 Table) [31]. This study was approved by the Griffith University Human Research Ethics Committee (reference 2017/951).

### Participants

Eligible individuals were those newly diagnosed with T2D who were participating in a 12-month observational study with no intervention (Diet after a Diagnosis of type 2 Diabetes: The 3D Study [32]). The 3D Study obtained information on demographic, physical, and social psychological factors of 225 individuals newly diagnosed with T2D [32]. While author RR was involved in collecting data for the 3D Study, this did not impact the current study as no prior relationships were knowingly established. Only the participants who consented to participate

in further research were included in the potential recruitment pool for the current study (see S1 Fig for the flow of recruitment). Purposive sampling from the 3D Study participant database was used to invite participation from people newly diagnosed with T2D who had consulted with a dietitian for dietary advice after their diagnosis on one or more occasions as it enabled the selection of information-rich cases [32].

Potential participants (n = 42) were invited by author RR via email, which included a participant information sheet and consent form and notification that the study was a component of the author's (RR) PhD research. The information and consent formed outlined that agreeing to participate was implied consent. Twelve individuals did not respond to any recruitment contacts, and one individual sent an email indicating declined participation. Two participants did not answer their interview call and were uncontactable afterwards. The final number of participating subjects was 21.

## Interview guide development

A semi-structured interview guide was developed using open-ended questions to stimulate discussion about the participant's retrospective experience of consulting with a dietitian (e.g., "Let's now focus on your time with the dietitian, can you tell me about that and your experiences.") and the influences on their dietary change (e.g., "What helped you to make the decision to make those [dietary] changes?") (see S2 Table). Formulation of the interview protocol was informed by previous work in this area [33], the expertise of the research team, and the overarching interview topics that guided the study. To enhance qualitative credibility [34], participants were prompted to recall specific situations where they had made changes to their diet just after their diagnosis, either self-enacted or as a result of consultation with a dietitian and to discuss whether changes were made before and after seeing their dietitian (e.g., "Now let's focus on your diet and foods choices after your T2D diagnosis. I'd like you now to tell me about your experiences after you've been diagnosed with T2D"). To provide a balanced view of their experience, the participants were asked to elaborate on aspects that were and were not helpful, including any affective responses to this. Further exploration into these self-enactment behaviours and their influences will be reported separately in a subsequent paper.

## Interview protocol

Participants undertook a single semi-structured telephone interview with author RR, a female PhD candidate and Accredited Practising Dietitian, with honours-level experience and postgraduate studies in qualitative research. Interview times were arranged via email and conducted at a time convenient to the participant. Author RR's PhD research is in behavioural science in dietetics. She was previously involved in the 3D Study data collection, which likely shaped her interpretation of the current study's data. Interviews lasted between 18 and 40 minutes (mean length 26 minutes). Each participant received an AU$20 gift-card to thank them for their participation.

At the beginning of each interview, participants were informed of the process and content of the interview and provided with a working definition of "dietary behaviours" as: ". . . your food choice, eating behaviours and habits, and specific dietary or nutrition intakes". Then the participants were asked to expand on how a dietitian influenced these dietary behaviours (i.e., probe: "*In what ways did your dietitian help you make dietary changes*?"). The interviewer summarised the discussion with each participant to ensure the qualitative validation of collected information [35] and invited each participant to modify or elaborate on this summary. Participants consented for the interviews to be audio recorded, using a digital voice recorder and transcribed verbatim for data analysis. To ensure participant anonymity, all interview

recordings were permanently deleted after being transcribed. Notes were made by the interviewer during each interview to explore relevant concepts further. These notes were not provided to participants. Transcription was initially in a Microsoft Word document and then imported into a Microsoft Excel file for data analysis.

### Analysis

A theoretical thematic analysis of transcripts was performed to facilitate the emergence of themes [35]. This method was selected as it is guided by existing theoretical concepts and the researchers' standpoint and disciplinary knowledge. The six phases of thematic analysis by Braun and Clark (2013) were carried out: data familiarisation through transcribing data, reading and rereading the data; generation of initial codes guided by the theoretical constructs of the analysis guide; identification of themes; review themes; and define and name the overarching themes [35].

The analysis was completed in the context of the theoretical framework used. An analysis guide, developed by author RR and cross-checked by author KH (a behavioural scientist), was based on integrating key psychological constructs from the IBC model [17]. The guide was used to understand concepts that underpin motivational (i.e., involving factors that influence the formation of a conscious intention to change dietary behaviours), volitional (i.e., involving factors that help transition the intention into the behaviour), and implicit (i.e., non-conscious thought processes that can impact behaviour spontaneously) processes that may influence the dietary behaviours of individuals diagnosed with T2D. However, some aspects of participants' narratives did not fit within the scope of these three broad psychological processes and other concepts, such as emotions and individual difference factors, [5, 16] were identified. Therefore, data coding was flexible to allow new codes that did not fit the broad processes of the IBC model to be added and categories and themes to be re-organised.

Coding and themes were reviewed by author KH (two coders in total) on 33% (n = 7) of the transcripts to identify and resolve any inconsistencies in coding and naming. As the data was collected, coded and analysed in an iterative process, recruitment ceased when the data no longer added anything new to the overall analysis [35]. The participants were not provided with the opportunity to review or comment on themes, nor did they provide feedback on the findings. Still, this learning was noted as a recommendation for future research. The authors reflected on the themes to ensure the concepts aligned theoretically to the motivational, volitional, and implicit processes of the IBC model, taking into account the additional themes that emerged outside of the model. The additional themes that emerged through participants' narratives aligned with the research question and have been reported in this paper.

### Results

Participant characteristics can be found in S3 Table. Participants included 21 adults (12 females and nine males), aged 36–75 years (mean age 61.38 years). The mean time from reported T2D diagnosis to interview was 20 months (range 16–24 months). The 3D observational study data indicated that participants reported having attended between 1 and 13 dietetic consultations from the time of their diagnosis to when interviewed for this study. The majority of participants (n = 14) reported having seen a dietitian one to four times. Six participants saw their dietitian five to seven times, and only one participant reportedly had 13 consultations with their dietitian. From the time of diagnosis, the participants reported to make contact with their dietitian within the following month, with one seeing a dietitian the day after their diagnosis and some a few weeks later. All participants said they were referred to see a dietitian through their GP, with 12 reporting using a dietitian through their General Practice

Management Plan or "care plan". It was not made clear whether the other nine participants used a care plan to see a dietitian. The mean BMI of all participants at the time of diagnosis was classified as obese (30.04 kg/m$^2$; SD = 4.56; range = 21.1 kg/m$^2$–41.9 kg/m$^2$).

Fifteen themes and 19 subthemes were identified from the theoretical thematic analysis (see Table 1 for representative quotes for each theme and subtheme). Fig 1 illustrates the interrelationship between the themes of each process. Five themes were classified as motivational processes, four themes as volitional processes, and two themes as implicit processes. Results were mostly similar across participants and fit the constructs and processes of the model. An additional four themes came through strongly in the data but did not distinctly fit the IBC model; these were classified as individual difference factors and emotions. For each quote, participant details are presented for their participant number (P#), sex (F = female; M = male) and age in years.

## Motivational processes

All participants discussed various motivating factors that enabled them to make positive changes to their diet. The motivation stemmed from five key points which are expressed in the motivational themes and included wanting to *improve their health status*, make changes for *themselves and others*, *making use of social networks*, have *the ability and responsibility to make changes*, and *post-diagnosis realisations*.

**The desire to improve health status.**   It was evident that many participants had a desire to improve their health status. Three subthemes of the desire to improve health status theme were *getting diabetes under control*; *reducing further complications*; and *not wanting to be on medication* (see quotes in Table 1: Section 3.1.1) After their diabetes diagnosis, the majority of participants discussed how they were more consciously aware of their health and, therefore, motivated to make the necessary changes recommended by their dietitian or general practitioner to get their diabetes under control (i.e., make healthy dietary changes, lose weight, exercise more). One participant reflected on her mother's experience with poorly managed T2D and expressed not wanting to go down the same path.

In the second subtheme *reduce further complications*, a few participants expressed that they did not want their T2D to 'worsen' to type 1 diabetes. Although this is not a physiologically accurate progression of T2D, it illustrates a willingness to improve health status and an understanding that insulin injections may be required if their diabetes is poorly controlled. In addition, this motivation to improve health status seemed driven by a desire to reduce the possibility of experiencing further health complications associated with T2D. Some participants discussed not wanting to be faced with the possibility of having toe amputations, Alzheimer's disease or dementia, whether or not they understood the absolute risk.

In the third subtheme *not wanting to be on medication*, some individuals discussed how they were motivated to make dietary changes to reduce their chances of requiring medication for their diabetes, which is a reality for many people with T2D. Overall, this theme highlighted that many participants expressed some desire to improve their health status and demonstrated a willingness to make changes to make this happen, which was evidenced by consulting with a dietitian and the dietary changes discussed.

**To do it for themselves and others.**   Motivation to change dietary behaviours were internally and externally driven, which underpins the psychological constructs of intrinsic and extrinsic motivation. Four subthemes related to doing it for themselves and others include: *enjoy a little bit more life*; *wanting to be present for family now and in the future; being a role model*; and *pressure to comply* (see quotes in Table 1: Section 3.1.2). After being diagnosed, some participants expressed a desire to enjoy more of life. Therefore, they recognised to need

**Table 1.** List of the 15 themes, accompanying subthemes and representative participant quotes, for a qualitative study exploring the decision-making processes of dietary behaviour change after a diagnosis of type 2 diabetes.

| Main theme | Subtheme | Representative Quotes |
|---|---|---|
| **3.1. Motivational processes** | | |
| *3.1.1. The desire to improve health status* | | |
| | Getting diabetes under control | *"That motivating factor for me, well as I said. . .I wanted to maintain a good level with my diabetes, and I didn't want it to get out of hand, you know, I've done the diet plan she* [dietitian] *told me and I was walking every day of the week and playing golf twice a week just to make sure I did have my diabetes, you know, fairly well under control"* (P18, M, 75) |
| | Reducing further complications | *"Well I don't want to end up with type 1 diabetes"* (P11, F, 56) *"My dad died of Alzheimer's and sort of saying that diabetes type things can link in there, and not 100% sure but there may be a link there"* (P11, F, 56) |
| | Not wanting to be on medication | *"Yeah well, I wasn't very happy with that* [the possibility of being on insulin], *and that just makes you work harder on not having to have it"* (P02, F, 69) |
| *3.1.2. To do it for themselves and others* | | |
| | Enjoy a little bit more life | *"So, I just came to the realisation that if I wanted to enjoy a little bit more life, I've gotta [sic] change. And the way I was going wasn't a good way to go"* (P19, M, 66) |
| | Wanting to be present for family now and in the future | *"Yes, and plus when you've got family, you know, kids and grandkids, and you think. . .when you've got that in your history, you really are more like, okay, we're just going to keep pushing at this"* (P09, F, 72) |
| | Being a good role model | *"I've got young grandchildren, so I think, you can't really drink soft drink and then tell the kids you can't have soft drinks, so, it was sort of I suppose trying to model for them, um, you know, what good dietary habits are"* (P08, F, 70) |
| | Pressure to comply | *"Probably being scared of my doctor. . .If I'd gone and seen her and I hadn't done the right thing she'd be really cross* [laughs]*"* (P11, F, 56) |
| *3.1.3. Making use of support networks* | | |
| | Simple and easy to understand dietary advice | *"She gave me a couple of websites to go and visit to look at menus and ways of changing, or replacing, swapping out foods to stuff that's good for you"* (P14, F, 50) |
| | Available and dependable support networks | *"She* [dietitian] *was also accessible by phone or email if I want, if I had any questions, she said look, if you, you know, something pops up and you're worried, email me, here's my email just so, so she was always very accessible which is, I think is important"* (P03, F, 46) |
| | Encouragement from others | *"I went back to see her* [dietitian], *and she was just over the moon, And she's just, very encouraging. You know she said, 'look you've done a fantastic job, and you're well motivated', and you know it's a win-win for you"* (P03, F, 46) |
| | Lack of social support | *"The problem you get in a household, is no one else wants to eat like that, you know what I mean, it gets really hard. If I could do it for everyone, it would make it so much easier, but you can't, you've still gotta [sic] cook for other people"* (P15, M, 54) |
| *3.1.4. Ability and responsibility to make change* | | |
| | Ability within themselves | *"Yeah, she would tell me alternatives. Like, instead of that, you can have that, and this and all the rest of it. It was, it made me think, you can really change, that you can eat the things you can but just moderately than anything else"* (P17, F, 75) |
| | Recognition of own responsibility | *"I've gotta try do something to control it* [diabetes] *rather than the other way around"* (P08, F, 70) |
| *3.1.5. Post diagnosis realisations* | | |

*(Continued)*

**Table 1.** (Continued)

| Main theme | Subtheme | Representative Quotes |
|---|---|---|
| | This is a wakeup call | "*You, you hear people saying, oh I have to lose weight I got to lose weight. But you'll never do it unless you're faced with something that gives you the motivation*" (P03, F, 46) |
| | Shock to the system | "*Yeah, it was a bit of a shock, you know, like well, what does this mean*" (P20, M, 52) |
| **3.2. Volitional processes** | | |
| *3.2.1. Knowing their limits* | | "*Rather than just yes I'll have my third helping of pasta because it's delicious, but, no, I'll have a small bit of that pasta and that will be it*" (P11, F, 56)<br>"*But you know you still get to eat it and you still get to enjoy it, but you can't pig out on it*" (P11, F, 56) |
| *3.2.2. Dealing with falling off the wagon* | | "*Now I feel that if I want to, I have a splurge occasionally, like go out for dinner and just eat normally, or whatever there and enjoy it, but the next morning, I get back on the wagon again*" (P19, M, 66) |
| *3.2.3. Learning how their body responds to food* | | "*Because you can see immediately how um carbohydrate related to your blood sugars, if you keep a food diary. And then you do your blood sugars six times a day before and after meals and between meals and all that sort of stuff- you can see just at a glance, exactly what's going on. And I learnt such a lot from that*" (P03, F, 46) |
| *3.2.4. Plans to meet the end goal* | | |
| | Watching their carbohydrates | "*He* [dietitian] *said, 'Yep, just any food label, in a nutshell, anything more than 15 grams of sugar per 100 grams don't touch it', and I thought perfect, that's what I needed to know*" (P12, F, 53) |
| | Preparation is key | "*I eat so much more healthier now, even like today, I don't buy my lunch, I take my lunch to work now. Fruit, nuts, good salads, all that type of stuff*" (P19, M, 66) |
| **3.3. Implicit processes** | | |
| *3.3.1. Unlearning habits* | | "*So I had to switch me brain on to remind me, that at 10 o'clock, even if I wasn't hungry, you still gotta [sic] have something to eat*" (P10, M, 64) |
| *3.3.2. Limit the availability* | | "*I don't keep you know, any crap in the house anymore, if you know what I mean, I don't, you know I can't open my pantry now and see you know chips and biscuits and that sort of stuff, because if it's not there, then you can't be tempted to eat them*" (P11, F, 56) |
| **3.4. Individual difference factors** | | |
| *3.4.1. Interpersonal style of consultation* | | "*She* [dietitian] *basically said it as it was, if you know what I mean, so, I like, probably a more direct approach*" (P19, M, 66) |
| *3.4.2. It is the type of person they are* | | "*I certainly didn't have feelings of anger or despair, I'm certainly not that type. You have to face situations*" (P12, F, 53) |
| **3.5. Emotions** | | |
| *3.5.1. Changing food can be emotional* | | |
| | Emotional response to food | "*I was a bit shocked 'cause [sic] I like fruit but, um it didn't matter*" (P02, F, 69) |
| | Missing the food they love | "*Well I miss eating lots of bread. . .but I, have to be very strict with myself because I don't want to get back to how I was before*" (P03, F, 46) |
| *3.5.2. Feelings of shock, worry, and no surprise* | | "*I felt a bit scared and then started doing research on how bad it is and what you can do, and then two, three months later, my sugars had dropped very considerably*" (P20, M, 52) |

to change their behaviours to do so. For example, participants discussed wanting to lose weight to make themselves feel better, experiment tasting different foods as they found a new pleasure in cooking, and have a sense of control over their diabetes. A few participants who had made

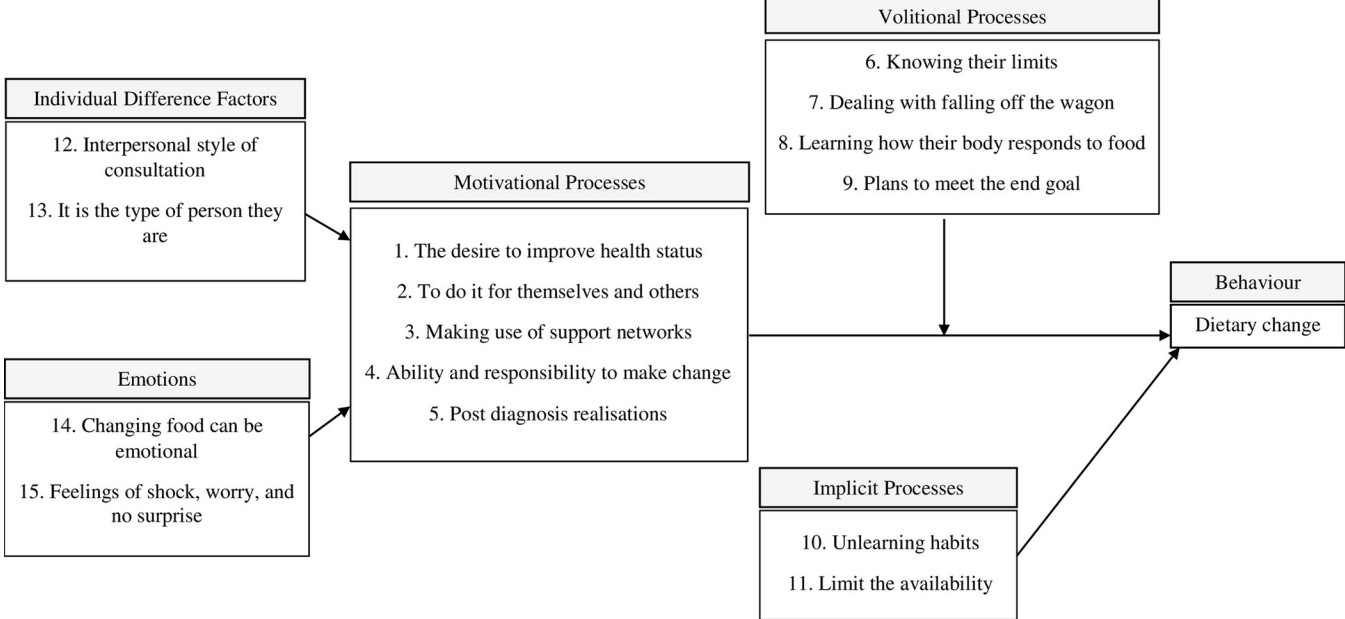

**Fig 1. Visual representation of the 15 themes identified in a qualitative study exploring the decision-making processes of dietary behaviour change after a diagnosis of type 2 diabetes guided by the motivational, volitional and implicit processes of the integrated behaviour change model [17].**

long-lasting dietary changes that resulted in clinical improvements in their diabetes, talked about their strong-mindedness to improve their health status to feel better. However, for one participant (P07, M, 73), a lack of internal motivation to make change was reflected in his dietary intake. Despite having the self-efficacy, knowledge, and support (wife) to make dietary choices recommended for T2D, including more than the mean number of dietitian visits (n = 7), it was clear that he was not intrinsically motivated to do so as he mentioned he was unable to adhere to long term dietary changes. The participant expressed how at first you have good intentions to make changes, so the desire and priority on health is strong; however, this then tapers off over time and "*you drift back to old habits again*" (P07, M, 73). In contrast, the second subtheme *wanting to be present for family now and in the future*, suggested others drive a motivation to change that. This more extrinsically driven motivation came through strongly as many participants talked about their families and the desire to be around for them, now and into the future.

In the third subtheme *being a good role model*, some participants discussed feeling responsible for maintaining their health to stay alive and be good role models for their children and grandchildren. A few further highlighted the important role they play in their family's lives and, therefore, the need to be a positive role model. Interestingly, some participants discussed the *pressure to comply* with the advice they received from health professionals, which embodied the fourth subtheme. Further, some participants discussed how they were advised to keep a food diary to discuss with their dietitian on the following visits. This type of monitoring from someone considered an 'expert' seemed to bring about a sense of obligation to change.

**Making use of support networks.** Almost all the participants discussed the role that social support played in their dietary intake and their motivation to change. Four subthemes of making use of valuable support networks were *simple*, *easy to understand dietary advice*; *available and dependable support networks*; *encouragement from others*; and *lack of social support* (see quotes in Table 1: Section 3.1.3). The support discussed by most participants was information-

giving from their dietitian or general practitioner, both about diabetes as a disease and diabetes dietary advice. The support typically occurred in the context of specific changes that needed to be made to control their diabetes. For most participants, receiving *simple*, *easy to understand dietary advice* on label reading, carbohydrate intake, and portion sizes were discussed as being very useful. One participant discussed how their dietitian talked about "*actual products to buy*" and "*looked at the whole matter holistically*" (P05) which he found useful. This participant had the 13 consultations which may have allowed the time to discuss such matters. A couple of other participants said their dietitians provided fact sheets on food groups and information on national dietary guidelines. Other useful advice included the information that dietitians provided around dietary habits that are 'good' and 'bad', websites to use for menus and recipe ideas, and education on diabetes and the dietary components related to it. However, a few participants mentioned their dietitian being reportedly too prescriptive with their dietary advice, as they expected more education rather than a "*do not eat this*" list. It was clear among these few participants that they felt unheard by their dietitians and the consultations lacked patient-centeredness. Overall, these experiences did not appear to hinder their ability to make dietary changes; however, it might have prompted reluctancy to seek dietetics advice in the future, as the few participants that reported this saw their dietitian one to two times.

Having *available and dependable support networks* such as diabetes groups (e.g., DES-MOND (self-management education course for people living with T2D)), food apps (Under Armour, Food Switch, and Calorie King), and encouraging family members was discussed in helping participants be motivated to change. A few participants also mentioned how their dietitian was readily available to call or email in times of need, which acted as a form of reassurance.

In the third subtheme, some participants discussed the importance of *encouragement from others*. This encouragement came from work friends or within their close circle, who commented positively on their progress, such as losing weight or adhering to their diet. Receiving this encouragement support in the form of positive feedback on the outcomes of their dietary changes from friends, dietitians, and other health professionals were discussed as positively influencing their desire to make dietary changes.

In contrast, some participants mentioned how a *lack of social support* hindered their ability to make the dietary changes needed for their diabetes control. For example, those in charge of preparing food for the household found it challenging to meet everyone's needs and preferences, and, consequently, their own dietary requirements took a lower priority. In some cases, while frustration was experienced initially, their families eventually became supportive of the dietary needs of the participants, highlighting the importance of social support for dietary change, *"It's much easier now that I'm only cooking one meal and everyone's just eating what I cook"* (P13, F, 36).

**Ability and responsibility to make dietary changes.** It was clear that many participants had some form of motivation to change, whether it was to improve their health status or reduce the risk of further complications. However, participants' narratives also highlighted the role that ability and responsibility played in their dietary changes. Two subthemes of ability and responsibility were *ability within themselves* and *recognition of own responsibility* (see quotes in Table 1: Section 3.1.4). Most of the participants discussed that combining a good support network alongside a high level of motivation meant that they felt it was within their ability to make changes. Some participants discussed how they felt a sense of responsibility to take ownership of their health and recognised that they are the one in control of their health and dietary choices. This motivation stemmed from wanting to do things for themselves. It was clear from these narratives that participants felt a need to "embrace" (P19, M, 66) their diabetes and take back control over it.

**Post diagnosis realisations.** Two sub-themes of post diagnosis realisations were *this is a wakeup call* and *shock to the system* (see quotes in Table 1: Section 3.1.5). For many participants, the diagnosis of T2D was the initial wakeup call they needed to motivate them into action to change their dietary and lifestyle behaviours. Some participants confessed to an ongoing denial regarding their dietary intake or weight issues, and that they were motivated to make changes after receiving a formal diagnosis of T2D. Related to T2D diagnosis as a driver for change, some participants expressed that they had not thought they were at risk of T2D. Thus, receiving the diagnosis came as a shock and motivated them to make dietary changes accordingly. This affective response is described in further detail in the emotion theme.

## Volitional processes

Through conscious decision making, some participants expressed how they overcame challenges and constraints when trying to change their dietary behaviours. The participants discussed their self-control, how they had coping mechanisms in place, practised self-monitoring, made plans or goals and consciously made dietary choices recommended for T2D. These four key points are expressed in the four volitional themes and two subthemes.

**Knowing their limits.** Through *knowing their limits* (see quotes in Table 1: Section 3.2.1), a few participants talked about the strategies they put in place to avoid certain temptations, which was considered as a form of self-control. These participants talked about their awareness of temptations, such as overindulging in chocolate or giving into sweet cravings, a love for pasta, and dining out, which was especially strong in the early stages after their diagnosis. Through their narratives, some participants demonstrated restraint by reducing portion sizes or actively thinking about their food choices and choosing 'diabetes friendly' alternatives. This sense of self-control over their dietary choices seems to stem from their real need to improve their diabetes. Additionally, by learning about what foods they should avoid (such as sugar-sweetened beverages, cakes, or chocolate), many participants expressed how they managed these constraints by changing the way they think about food. For some, they were aware of their cravings and stated that, in the past, they would have consumed their desired food without much thought. However, now, they described being more conscious of these cues and therefore overcame their cravings through pre-emptive means.

**Dealing with falling off the wagon.** Almost half of the participants talked about how they dealt with minor setbacks in their diet or around 'cheating' on their recommended diets in the theme *dealing with falling off the wagon* (see quotes in Table 1: Section 3.2.2). This theme aligns with the psychological construct of coping self-efficacy, which represents a person's optimistic beliefs about their capability to cope with barriers that arise during the period of behavioural maintenance. Some participants discussed how they would make sure they went for a walk after indulging on a piece of cake or stated they would make sure they got "*back on track*" (P06, F, 61) with their diet the following day. These participants expressed their understanding that not all diets can be adhered to 100% of the time, and the importance of incorporating additional measures to counter any 'slip-ups', such as extra walking to counter food indulgences. Through a few participant narratives, it seemed the dietitians played a role in helping stay on top of any advised dietary recommendations. Being provided with reassurance, and frequent "don't stress" (P01, F, 73) comments made by the dietitians, allowed participants to feel that they could adjust and cope with dietary recommendations and feel confident getting back on track following a setback.

**Learning how their body responds to food.** Just over half of the participants talked about how they regularly monitored their blood glucose levels or kept a food diary to monitor their dietary intake when they were first diagnosed with T2D. This theme underpins the

psychological construct of self-monitoring (see quotes in Table 1: Section 3.2.3). They mentioned how this enabled them to learn how their body responded to certain carbohydrate-type food (such as bread, alcohol, fruit) which then influenced their food choices. Some participants either avoided certain foods or used their readings to reassure themselves that what they were consuming was safe regarding their sugar levels. Some participants were encouraged to keep a food diary, to allow their dietitian to know what they were eating, and to increase self-awareness of dietary intake through the process of reflecting on their intake, seeing the physical proof, and then making change. A few participants mentioned how the process of keeping a food diary and discussing it with their dietitian enabled them to learn about what foods affected their body and where adjustments then needed to be made in their diet as recommended by their dietitian.

**Plans to meet the end goal.**   Several participants talked about practical tips and strategies they put in place that helped them adhere to dietary recommendations and enabled them to make dietary changes. Two subthemes of plans to meet the end goal were *watching their carbohydrates* and *preparation is key* (see quotes in Table 1: Section 3.2.4). The first subtheme *watching their carbohydrates* was quite specific and highlighted the relationship between carbohydrates and diabetes. From their consultations with dietitians, the participants mainly focused on carbohydrate-type foods, which most participants referred to as "carbs" and "sugars". In relation to the dietary advice received, one participant reported their dietitian "*said it was all based on carbs*, *like intake of carbs*" (P06, F, 61) and another said their dietitian went through "*carbs and calories*" and "*low-carb style diets*" (P19, M, 66). The carbohydrate-type foods commonly reported to be removed from the participants' diets included added sugars, chocolate, cakes, soft drinks, bread (particularly white) and pasta. Some participants said they aimed to have a certain amount of carbohydrates either per meal such as 40 grams, or per day such as 120 grams or having 20% of their total energy intake from carbohydrates. One participant (P10, M, 64) discussed he now makes sure he eats something before drinking alcohol, to slow down the absorption of alcohol, so it has less effect on this diabetes. Having these measurable plans appeared to help the participants manage their dietary intake for their diabetes and showed the emphasis participants placed on carbohydrate-type foods over other nutrients.

In the second subtheme *preparation is key*, the participants talked about how they had strategies to ensure they were consistent with their diet and did not deviate too much from their goals. One participant said she packs a small portion of almonds in her purse to stop from over-snacking and to know her limit of what she can eat. One participant said he cooks in bulk and freezes food for lunches for the week to ensure his intake was consistent and ease the meal preparation burden. Another participant said he now packs his lunch instead of buying it at work. From these narratives, meal planning and awareness of recommended foods allowed participants to stick to recommendations and make dietary changes.

## Implicit processes

Implicit processes are suggested to represent more automatic, non-conscious decision making, which requires little conscious effort in undertaking a behaviour. The themes discussed by some of the participants that reflect this process involve unlearning habits and limiting the availability of food in their physical surroundings.

**Unlearning habits.**   Some participants talked about the habits they had to unlearn to make dietary changes for T2D in the theme *unlearning habits* (see quotes in Table 1: Section 3.3.1). It is recognised that the process of unlearning habits is a deliberative process. However, this theme captured the automatic processes and habits that participants identified they needed to change, and then provided examples of how that was carried out. For example, one participant,

who was previously in the army, talked about how that environment had helped to create some of his 'bad' eating habits over time, such as eating too quickly, and that this behaviour was still ingrained. He discussed how he wanted to break this habit of "*fast eating*" (P05, M, 67) and actively sought his dietitian's help to manage this. This participant had more visits with his dietitian than the mean participants of this study, which may have allowed ongoing support for habit reversal. Another participant talked about how learning to eat more regular meals consistently throughout the day was challenging as he usually did not eat in the morning or late afternoon. Another participant talked about how he is an "*old fashioned eater*" (P18, M, 75), eating a meal consisting of meat and three vegetables a day. After discussion with his dietitian, he had come to recognise that he needed to include more salads.

**Limit the availability.**   In the second theme *limit the availability* (see quotes in Table 1: Section 3.3.2), a few participants talked about how their physical surroundings facilitated dietary behaviours unfavourable for T2D. Having chocolate readily available in the house or having large bowl sizes that resulted in overeating were discussed. One participant (P12, F, 53) reported that she overconsumed convenience food when driving long distances for work and was in locations with limited availability of appropriate food. She had restructured her environment (her car) and installed a fridge which enabled her to pack her lunch and choose food options that align with T2D recommendations. One participant who worked away at sea had limited control over the food provided on site, therefore his agency to choose food for his diabetes was restricted beyond his control.

**Individual difference factors.**   Two themes were identified for individual difference factors which suggested additional themes to those of the IBC model. Some participants' responses to their diagnosis reflected their interpersonal style or the possibility that of the dietitian. The participants also talked about the type of people they are, which was reflected in how they talked about their changes.

**Interpersonal style of consultation.**   The first theme of individual difference factors highlights the preferences the participants had for their dietetic consultations (see quotes in Table 1: Section 3.4.1). For example, a few participants of this study, in particular men, talked about how they preferred a direct approach from their practitioner. One participant (P05, M, 67) mentioned that he enjoyed his dietitian's ongoing support, having gone every six weeks to stay updated. In contrast, another male felt he got the most out of his consultations because he only had two, therefore being focused and efficient. A couple of participants, who were female, preferred a less directed and counselling style. One stated how purely having someone to talk to and counsel her was most beneficial "*I think in him* [dietitian] *spending the time with me more on a counselling situation, I think the penny dropped*" (P08, F, 70) which, in turn, helped her make positive dietary changes. When talking directly about their dietitian, a few participants described their dietitian as "*pragmatic*" (P04, M, 55), "*gentle and unassuming*" (P07, M, 73), "*professional and thorough in their approach*" (P05, M, 67), or "*very factual and easy to relate to*" (P21, M, 68). These participants went on further to discuss positive changes they made to their diet and lifestyles. On the other hand, one participant saw a dietitian who they felt was "*inappropriately assertive*" (P20, M, 52). He felt judged, which resulted in a negative attitude towards his dietitian and hesitation to see another dietitian.

**It is the type of person they are.**   In the second theme of individual difference factors, some participants talked about the type of person they are in response to their diagnosis or how they responded to situations (see quotes in Table 1: Section 3.4.2). When talking about making changes, one participant said he does things "*to the nth degree*" (P11, F, 56). Another described himself not to have feelings of anger or despair when talking about how they responded to their diagnosis. One man said he is a "*stick in the mud type of person*" (P18, M, 75), in reference to his dietary intake deviating little from day to day. In terms of being able to

make changes, one participant said, "*When I set my mind to it, I've got a pretty strong will*" (P19, M, 66).

**Emotions.** A further theme clearly expressed through participants' narratives and additional to the IBC model was emotions. Two themes were identified: their emotional connection to food or emotions from receiving a T2D diagnosis. The emotions expressed seemed to influence individuals' motivation for change and how they changed their dietary behaviour. The emotional responses described by participants appeared to stem from the dietary changes needed to be made and the diagnosis of diabetes itself. Participant responses did not appear to suggest whether their dietitian interaction elicited positive or negative emotional responses.

**Changing food can be emotional.** Two subthemes fall under this overarching theme, *emotional response to change* and *missing the food they love* (see quotes in Table 1: Section 3.5.1). Many participants highlighted various emotions when discussing the dietary changes they needed to make when they were initially diagnosed, and afterwards. In the first subtheme, some participants said they were shocked at what they could no longer eat when they were first diagnosed, especially relating to fruit. One participant discussed how she was surprised that her dietitian said, "*the things that you are allowed to eat doesn't mean that you can eat heaps of them*" (P12, F, 53) such as strawberries. Those participants who said they felt guilty after eating something they should not (such as birthday cake) reverted to their coping mechanisms and reminded themselves not to stress and get back on track the next day.

The second subtheme *missing the food they love* explains the emotions felt in the period following T2D diagnosis for participants. For some, they expressed frustration in social settings or felt "*deprived of something out of the diet range*" (P02, F, 69) (such as having cake with coffee). Others missed food from the carbohydrate group such as mashed potatoes, bread, and fruit. However, the few participants who said they missed certain foods or felt "deprived", mentioned how certain amounts of those foods were not suitable for diabetes. Therefore, they no longer ate them or ate in moderation (for example, only eating half a banana, eating three dates when missing sweet food). On the other hand, a few participants who did not express a strong connection to food appeared to find it easier to change. Some participants said they did not miss certain foods (such as wine or coffee) or were not worried about cutting out sugar and bread.

**Feelings of shock, worry, and no surprise.** Many participants talked about emotions brought on by receiving the T2D a diagnosis in the theme *feelings of shock, worry, and no surprise* (see quotes in Table 1: Section 3.5.2). For many, it was unclear whether the emotions felt had a direct impact on their change processes. However, some participants discussed the dietary changes made because of these emotions. A few of the participants were shocked from what seemed to be an unexpected diagnosis, saying they were "*reasonably fit*" (P07, M, 73) did not believe they would get diabetes. Some even stated their frustration that their health had declined. This initial shock was described as a powerful emotive response to instigating a change in dietary behaviours such as immediately cutting out added sugars such as sugar in coffee.

However, a couple of participants stated they were not surprised with their diagnosis, with one saying, "*diabetes is rife through our family. . .I wasn't surprised at all*" (P12, F, 53). These participants talked about making dietary change with ease, having a "deal with it" mindset. Some participants were aware of and acknowledged the emotions felt throughout the early stages of their diagnosis. However, when drawing on the motivations and willingness to control their T2D, this initial emotional response did not appear to have a strong, if any, influence on dietary intake.

## Discussion

This research employed a qualitative methodology to examine decision making processes underpinning dietary behaviour change post a diagnosis of T2D and the influence of dietetic consultations. This understanding was achieved by theoretically analysing participant narratives using the IBC model as an overarching framework [17]. To our knowledge, this study is the first to apply an integrated model of behaviour to explore and understand the decision-making processes in relation to dietary change in this demographic of adults. Participant experiences in the group largely corresponded with the motivational, volitional, and implicit processes of the IBC model; however, these were often inter-related with motivational influences emerging as a key feature enabling the other processes. The data also revealed other influencing factors not identified in the IBC model, such as individual difference factors and emotions, that influenced dietary behaviour change. Below we present a discussion of the five key themes in this data under the headings: motivational processes, volitional processes, implicit processes, individual difference factors, and emotions.

### Motivational processes

Some participants described their diagnosis as a *wakeup call*, which may suggest that a health-risk event could be the impetus needed to drive someone into action; in this context to change dietary behaviour. Findings from the English Longitudinal Study of Ageing found limited evidence to suggest that a T2D diagnosis encourages behaviour change other than for quitting smoking [36]. However, T2D could be suggested to be a 'teachable moment', which McBride, Emmons, and Lipkus [37] has defined to occur when individuals adopt risk-reducing behaviours following a health-related event (i.e., T2D diagnosis). This suggestion has been supported in other studies exploring cognitive, emotional, and behavioural changes following a diagnosis of T2D [38]. Such motivational processes can also be understood by reference to constructs that underpin the health belief model [39]. The model postulates that individuals engage in certain behaviours (i.e., dietary behaviour change) because they hold certain beliefs and perceptions regarding their susceptibly and severity of an illness (i.e., T2D) [39]. The theory also proposes cues to action as a precursor of behaviour change, and that the diagnosis of T2D may elicit this cue response. A mixed-method study of adults diagnosed with prediabetes found that some individuals would have made changes earlier, had they known the seriousness of T2D or been referred to a health professional like a dietitian at the time of their prediabetes diagnosis [40]. Addressing the seriousness of T2D well before a diagnosis could influence the likelihood of individuals adopting positive changes and delaying, or reversing, the onset of T2D.

Participants' beliefs surrounding their *ability within themselves* to change their dietary behaviours regularly featured in the narratives with several different themes intertwined. This ability highlighted the importance of self-efficacy for dietary change in individuals with T2D. In the health behaviour change literature, an individual's ability and confidence (i.e., self-efficacy) toward performing a target behaviour is a strong predictor of actual behavioural performance [21] and the participants expressed this in this study. Previous research has found a positive relationship between social support, self-efficacy, and adults' psychological well-being with T2D [41]. This finding concurs with a cross-sectional study demonstrating that high levels of self-efficacy were associated with diet adherence in adults with T2D [42]. Our results are consistent with these accounts in illustrating that self-efficacy beliefs can shape the dietary changes in individuals with T2D and highlight the importance of enhancing one's ability and confidence for change.

According to theories of social cognition, humans are social beings and can be heavily influenced to make decisions based on social relatedness and norms [19]. Such influences were identified in the theme *making use of their social networks*. Social support's benefits on enabling behaviour change have been well established in the health literature [43, 44] and social cognition models highlight the motivational influence that social support has on intentional behaviours [19]. Social support may also facilitate an individual's basic psychological need of relatedness, which is one of the central components of self-determination theory [19]. Our findings support this, as those participants who made use of their social networks and felt a sense of connectedness to their family or dietitian also reported making positive changes to their diet. Individuals with T2D often require a high level of support given the complexity of the disease [4], and this support can be received from a range of sources. Our findings are consistent with previous research that has found social support to positively influence a person's ability to cope with their disease and improve adherence to treatment [44]. Other research has indicated that social support can have a positive or negative influence on diabetes management. For example, a systematic review of observational and intervention studies exploring the impact of social support in adults with T2D highlighted how family members who were perceived to be overprotective and overwhelming could negatively impact a person's ability to self-manage their diabetes [45]. Although these characteristics were not explicitly mentioned among the current study participants, a lack of support from family members was described as a barrier to change. Interestingly, a study by Oftedal [43] found that adults with T2D experienced demotivation of their diabetes management through emotional support perceived as being non-constructive. In contrast, our study participants expressed emotional support to be constructive and positive to their overall diagnosis and experience with dietary change. These results demonstrate the type of support individuals might receive and provides insight into the potential impact that social support can have on motivation levels.

## Volitional processes

It has been suggested that individuals who make conscious and purposeful plans that are specific and measurable may be more inclined to achieve their goals [17]. In this study, the data revealed that participants often made plans that incorporated specific detail to have strategies in place to manage their food choice. In a longitudinal study of undergraduate students, action planning has been shown to have positive effects on dietary behaviour, in particular fruit and vegetable intake [46]. Considering adults with T2D have been found to fall short of the recommended intakes of foods such as fruits and vegetables [3], dietitians and other health care providers should encourage their patients to create detailed action plans to adopt and support recommended dietary behaviours. It was evident that most participants' recollection of dietary changes was focused on carbohydrate-type foods. However, dietary guidelines and T2D recommendations state that other nutrients are important for T2D management, such as protein and total energy intake [47, 48]. This apparent lack of reporting of other key nutrients suggests that individuals primarily relate T2D management with carbohydrates. It is important that dietitians and other health professionals involved in diabetes care educate patients to understand the appropriate dietary recommendations for T2D and address common ambivalences or lack of importance on other foods.

Other volitional determinants of dietary behaviour change identified in this study included self-monitoring. Our study suggests that individuals who say they are motivated to control their diabetes also report practising self-monitoring (i.e., food diary, blood glucose monitoring) as part of their diabetes management. These findings support the theoretical constructs in the volitional phase of the health action process approach [21] which posits planning and

action control (i.e., conscious monitoring and evaluation of actions against a desired behavioural standard) as helping to bridge the intention-behaviour gap by mediating the effect of intentions to actual behaviour [49]. Such self-regulatory practices are evidenced to enhance clinical outcomes [50] and should be a fundamental aspect of dietitians' nutrition care.

Within the volitional processes leading to a change of dietary behaviour, our participants highlighted the importance of self-control and *knowing their limits* when dealing with food temptations and having the capacity to manage challenges and setbacks to their diet. Self-control can refer to the capacity and ability an individual has in dealing with temptations, impulses, and habitual responses [51]; however, there are potential limitations on the sustainability of self-control over an extended period. Strength models of self-control show that a person's ability to have self-control on one occasion may weaken their ability on the next [51]. These findings suggest that individuals who are in the process of making multiple dietary changes should be mindful that a continuous effortful inhibition of impulses may, over time, be detrimental to their dietary goals if long term change is solely dependent on having self-control.

The literature on coping self-efficacy indicates that having the capabilities to manage challenges or external stressors is important in predicting change [39]. Coping self-efficacy allows individuals to actively approach challenging situations in a positive way and not be over-reliant on self-control [39]. This finding was expressed through *dealing with falling off the wagon*. Schwarzer and Renner [39] explored the relationship between action self-efficacy and coping self-efficacy. They found that individuals who report higher coping self-efficacy levels also reported better nutrition behaviours [39]. From this, the two psychological constructs of self-control and coping self-efficacy may lend well to each other. Dietitians should consider these constructs when addressing dietary behaviour change as diabetes management often requires a change in lifestyle behaviour and the need for self-regulatory matters like monitoring blood glucose.

## Implicit processes

The habitual nature of dietary intake and the relationship between poor dietary intake and health-related illness pose a real challenge to reverse such habits [52]. The types of habits derived from our participants' narratives have shown consistencies with previous research on food habits associated with T2D which relate to basic eating practices, dining out, meal planning, and carbohydrate/vegetable intake strategies [53]. Previous research on habits in health-related behaviours [52] and physical activity [15] indicates both automatic and conscious processes involved in food-related habits and associated behaviours. Some participants described engaging in some dietary behaviour without conscious thought and therefore, were done automatically. Such automatic behaviour aligns with the idea of habit formation, where a habit is considered "*a phenomenon whereby behaviour is prompted automatically by situational cues, as a result of learned cue-behaviour associations*" [52]. Some participants acknowledged the need to reverse and 'unlearn' these automatic processes to change their diet. *Unlearning habits* can be an arduous task given the processes involved in creating them [52]. Therefore, unlearning habits requires effortful self-control to practice and monitor behaviour progress from the individual [52]. Dietitians' dietary advice play an important role in supporting these changes. Dietitians can utilise behaviour change techniques to encourage patients to adopt self-monitoring practices and to prompt rehearsal of alternative behaviours to replace the unwanted habit [54].

The participants discussed *limiting the availability* of food that is unfavourable for T2D in their home as a way of 'restructuring their physical environment' to help influence a positive change in their dietary behaviour. This finding demonstrates the potential importance of

using such strategies to create barriers to unwanted behaviour. By reducing exposure to cues of the behaviour, the participants in our study said they could make food choices recommended for T2D. Research suggests it is easier to control one's behaviour when a less conscious effort is needed [55]. Restructuring the physical environment, for example by limiting the availability of non-recommended food in the house, may make it easier to control the temptation to engage in unwanted behaviour (such as eating chocolate). This restructuring means less conscious effort is required to control the impulse (because there is no readily available chocolate to eat).

While these changes were reported on an individual level, it is important to recognise that there are structural factors that play a role in food choice and dietary intake which are influenced socially and through obesogenic environments [56–58]. Some participants dined out due to convenience and accessibility of restaurants, which supports the literature on the role that environmental cues have on dietary behaviours [56]. Research on food temptations found environmental cues to influence both healthy and unhealthy eating choices [59]. Furthermore, the interplay of portion sizes, the built environment, change in work lives, technology, and food advertising within an 'obesogenic environment' [60] can also influence an individuals' motivation to eat out or choose certain foods regardless of their usual eating behaviours. These findings add to the body of evidence that the control an individual has over their environment may be limited and that the environmental cues play a large role in food choices whether individuals are aware or not [56]. These factors relating to obesogenic environments were only mentioned by one participant and the lack of direct responses around this topic may have been due to the focus of the interview protocol on individual factors. From a sociological perspective, it is important to acknowledge the structural variables that influence individual choice and the interplay with agency. Although not explicitly discussed by the participants, there may be structural limitations affecting their dietary choices that are beyond their awareness or control. Such factors may include financial ability to seek out a dietitian and purchase certain food products, as well as accessibility to services or living conditions [57, 58]. The interplay of agency and structure may be a limitation of the IBC model and future research may wish to consider these limitations when using the model from a behavioural science perspective.

## Individual difference factors

Individual difference factors, which distinguish people from one another, such as individuality and personality, were also important in participants' dietary change [61]. Our study highlighted that the interpersonal style of the dietitian perceived by the participant and their experience of the dietetic consultation had an influence on their motivation to make changes. Our findings are supported by Ball and colleagues [28] that found patients desire more individualised care from their dietitians. Our study's findings support that of Chen and colleagues [62] that the management of adults with T2D should address physiological, social, and psychological needs as each individual manages their diabetes care through their personal health experiences.

Individual difference factors are vital in providing a patient-centred level of care; therefore, dietitians and other nutrition care providers should be aware of these influential factors when trying to understand an individual's decision-making and the change process. Research in health interventions and behaviour change techniques has highlighted that the interventionist's interpersonal style is an important aspect when encouraging behaviour change [23]. However, Hagger and Hardcastle [23] raised an important point that interpersonal styles have not been, yet should be, distinctly included within commonly applied behaviour change technique

taxonomies. Although Michie and colleagues [63] distinguished between behaviour change techniques and 'aspects of the interaction' that are important for the delivery of behavioural support, our study adds further support to the arguments put forward by Hagger and Hardcastle. Our recent paper on self-enacted dietary behaviour change furthered the literature on behaviour change techniques by closely examining the techniques reported by the same participants of this study [64]. The study found a range of techniques that are important for decision-making processes that should be taken into consideration by dietitians, such as goal setting, problem solving and social support which align with the themes presented in the current paper [64].

Other individual difference factors revealed in the current data related to personality types and inherent traits. Personality dimensions (such as openness, conscientiousness, extraversion, neuroticism, and agreeableness [61]), have been explored to understand whether any associations between personality and dietary intake exist [65]. Research has shown that exhibiting more openness and conscientiousness traits are associated with healthier food intake and compliance with dietary recommendations [65]. It could be suggested that the participants in this study showed some levels of openness and conscientiousness as the majority were receptive of dietary advice, displayed creativity in food preparation, and had goal-directed behaviours to manage their diabetes [66]. These findings emphasise the importance of dietetic consultations being individualised whilst considering personality traits on health behaviours. The personality traits of dietitians have also been explored and indicate that there are notable differences between dietitians and the general public, and dietitians working in different settings [67, 68]. There is an opportunity for future work to explore further the impact of personality trait combinations in patients and dietitians on dietary behaviour change, especially in individual consultations that aim to support healthy eating.

## Emotion

Our study also highlighted that adults who receive a T2D diagnosis express emotional responses to their diagnosis and dietetic advice. Some of these emotions are supported by cognitive theories of emotion [69], as it was discussed that the T2D diagnosis itself triggered feelings of shock, guilt, and fear as motivating them to dietary change. These emotions are commonly associated with the emotional reaction of a T2D diagnosis [70]. As discussed above, a T2D diagnosis has been found to be a 'teachable moment', triggering cognitive, emotional, and behavioural responses which, in turn, affect decision making [38].

Much of the literature of food and emotion have explored the ways emotion affects eating (increased or decreased intake, cravings, appetite) and triggers for eating (comfort, lack of motivation, stressed, boredom, fear) [71]. These responses were found in some participant narratives who, for example, expressed feelings of 'nostalgia' by reminiscing on what they used to enjoy eating however can no longer have, a common emotional association to food [72]. It is perhaps unsurprising that these participants felt a sense of loss and were missing certain foods, as the foods they reported to have removed from their diet, like cake and chocolate, are known to be enjoyed more and with high taste appeal [72].

In regards to understanding the use of emotion for behaviour change, tailoring interventions to elicit an emotional response has been used to promote behaviour change [73]. Although the relationship between emotion on behaviour is well established, and given its influence on cognitive processes [74], there are few theories of behaviour change that incorporate emotion and affective constructs to help explain or predict behaviour [18, 19, 49]. Dietary interventions in the primary care setting are reported to use well-known behaviour change theories and models such as the social cognitive theory, transtheoretical model, and health

belief model which tend to focus on the social cognitions and volitional aspects of behaviours [25]. Our study's findings add to the body of evidence that emotions play an integral part in people's dietary intake and should be considered in behaviour change models.

## Strengths and limitations

Adults with T2D represent cases commonly encountered by dietitians in the primary care setting, and our findings are useful to those delivering nutrition interventions for adults with T2D. The participants in this study represent a population of adults who used early intervention (i.e., seeing a dietitian once newly diagnosed) to manage their diabetes. Many of these participants reported positive outcomes in dietary change, improved glycaemic control, and a better understanding of diabetes. Although the analysis was bound to the theoretical constructs of the IBC model, the model integrates key psychological constructs and processes which allowed for a deeper and more comprehensive understanding of health behaviours [17]. The analysis also allowed for themes to emerge beyond the IBC model. However, these links, including individual differences and emotions, did not come through as strongly in the data than prominent social cognition influences and, therefore, it is important that future research continue to explore these links.

The current study has some limitations. It is important to consider the possible bias in participants' responses. Although the study recruited n = 21 participants, which may be considered small, it should be noted that through the iterative process of data collection and analysis, the recruitment ceased when data and coding no longer added anything new. The individuals in this study were initially motivated to change their dietary behaviour and were willing to seek dietetic help. We also acknowledge the possibility of social desirability affecting responses. The interviewer was a dietitian, and given this status, participants might have felt a need to answer in a way they deem to be more socially acceptable than would be their 'true' response [75]. Future research can explore the lack of mention of financial difficulty amongst the participants, which may reflect a more affluent profile of participants as the study took place in a developed country with a high-income economy. Our data did not capture the processes of assimilating a diabetes diagnosis and how this might change over time for an individual, including how this understanding might impact on their dietary changes and diabetes health outcomes. Further research can explore the stages of understanding a diagnosis and the behaviours experienced by the same person to understand this process.

Finally, this paper presents major themes distinguished by the three processes of the IBC model; however, we acknowledge that many constructs within these processes are interlinked and lend to each other rather than being discrete as presented in this paper. Future research should explore the interconnections and overlap of themes, as well as the exploring whether the interaction between the dietitian and patient may elicit emotional responses. The interplay of agency and structure were not strongly represented in the IBC model which may limit the use of the model from a sociological perspective. As the participants' mood were not captured in this study, further research could explore the influence that mood and feelings associated with a type 2 diabetes diagnosis has on dietary intake and changes given the literature on emotion and decision making [74]. Our study did not capture the perspectives of young adults. Future research could explore this demographic as it may present different data compared to an older population. Weight stigmatisation and body image were not strongly identified in our participants. Given the known associations with adverse physiological and psychological health outcomes [76], future research can also explore the influence of body image, weight loss and diabetes. The quality of the dietitian consultations was unable to be captured in this study, which presents an avenue for future research that specifically asks questions relating to the quality practice standard outlined by the Academy Quality Management Committee [77].

## Conclusion

The use of behavioural theories to analyse dietary behaviours enabled a deeper understanding of the psychological processes of decision making in adults newly diagnosed with T2D. This study supported findings from the literature regarding the motivational processes involved in dietary behaviour change, as well as key volitional and implicit processes. The findings support the IBC model's use to understand dietary behaviour change in adults newly diagnosed with T2D; however, it also extends the model by addressing emotion and individual difference factors on dietary change. Moreover, this study's findings provide an evidence base for those providing nutrition care to expand their understanding of behaviour change science in future recommendations for dietary interventions to prevent disease and promote health, especially during the early stages of T2D diagnosis. Dietitians and other health professional providing nutrition care need to possess skills in understanding behaviour change to provide optimal patient-centred care. More research could empirically test the relationships between these psychological processes and dietary change in this population of adults to further the evidence base.

## Supporting information

**S1 Table. Consolidated criteria for reporting qualitative studies (COREQ): 32-item checklist for a qualitative study exploring the decision-making processes of dietary behaviour change after a diagnosis of type 2 diabetes (1).**
(DOCX)

**S2 Table. Overarching topics that guided the development of the interview questions for a qualitative study exploring the decision-making processes of dietary behaviour change after a diagnosis of type 2 diabetes.**
(DOCX)

**S3 Table. Characteristics and demographics of participants in a qualitative study exploring the decision-making processes of dietary behaviour change after a diagnosis of type 2 diabetes.** [a]BMI class classification according to WHO: [78]. [b]BMI class measurement taken at end of 3D-Study. [c]ARIA = Accessibility/Remoteness Index of Australia. Outlines the participants remoteness in Australia based on postcode [79].
(DOCX)

**S1 Fig. Flow of participants from 3D-Study to current study for a qualitative study exploring the decision-making processes of dietary behaviour change after a diagnosis of type 2 diabetes.**
(DOCX)

**S1 Dataset.**
(XLSX)

## Acknowledgments

3D Study participants for their ongoing contribution to research.

## Author Contributions

**Conceptualization:** Roshan R. Rigby, Lauren T. Williams, Lana J. Mitchell, Lauren Ball, Kyra Hamilton.

**Data curation:** Roshan R. Rigby.

**Formal analysis:** Roshan R. Rigby, Kyra Hamilton.

**Project administration:** Roshan R. Rigby.

**Supervision:** Lauren T. Williams, Lana J. Mitchell, Lauren Ball, Kyra Hamilton.

**Writing – original draft:** Roshan R. Rigby.

**Writing – review & editing:** Lauren T. Williams, Lana J. Mitchell, Lauren Ball, Kyra Hamilton.

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
