## [Decision Letter · Decision Letter 0]

28 Dec 2021

PONE-D-21-23420Understanding dietary behaviour change after a diagnosis of diabetes: a qualitative investigation of adults with type 2 diabetes.PLOS ONE

Dear Dr. Rigby,

Thank you for submitting your manuscript to PLOS ONE. After careful consideration, we feel that it has merit but does not fully meet PLOS ONE’s publication criteria as it currently stands. Therefore, we invite you to submit a revised version of the manuscript that addresses the points raised during the review process.

Please revise your manuscript according to the reviewer's comments.  Please submit your revised manuscript by Feb 11 2022 11:59PM. If you will need more time than this to complete your revisions, please reply to this message or contact the journal office at plosone@plos.org. Please include the following items when submitting your revised manuscript:A rebuttal letter that responds to each point raised by the academic editor and reviewer(s). You should upload this letter as a separate file labeled 'Response to Reviewers'.A marked-up copy of your manuscript that highlights changes made to the original version. You should upload this as a separate file labeled 'Revised Manuscript with Track Changes'.An unmarked version of your revised paper without tracked changes. You should upload this as a separate file labeled 'Manuscript'.

We look forward to receiving your revised manuscript.

Kind regards,

Olayinka O Shiyanbola

Academic Editor

PLOS ONE

Journal Requirements:

2. Peer review at PLOS ONE is not double-blinded (https://journals.plos.org/plosone/s/editorial-and-peer-review-process). For this reason, authors should include in the revised manuscript all the information removed for blind review.

Reviewers' comments:

Reviewer's Responses to Questions

**Comments to the Author**

1. Is the manuscript technically sound, and do the data support the conclusions?

Reviewer #1: Yes

Reviewer #2: Yes

2. Has the statistical analysis been performed appropriately and rigorously? 

Reviewer #1: N/A

Reviewer #2: N/A

3. Have the authors made all data underlying the findings in their manuscript fully available?

Reviewer #1: Yes

Reviewer #2: Yes

4. Is the manuscript presented in an intelligible fashion and written in standard English?

Reviewer #1: Yes

Reviewer #2: Yes

5. Review Comments to the Author

Reviewer #1: The theme of the article is extremely relevant to health issues, especially at a time when the Covid-19 pandemic, for obvious reasons, occupied the concern of both researchers and the general population.

Understanding in depth how patients deal with changes in their lifestyle and diet can help to overcome the difficulties they experience.

Dietary re-education and an individual-centred approach constitute the main strategies of the Australian health system which, unfortunately, are still not enough. Understanding the reasons for this limitation is very important to improve approaches with patients.

The text presents a theoretical model about the motivation to change behavior and also points out its limitations, especially with regard to personal and unconscious aspects.

The qualitative approach and in-depth exploration of collected data is an appropriate way to develop new models and, consequently, more adequate ways to guide health professionals and their patients.

At the end of line 169, the final number of participating subjects can be indicated.

Line 202 - review dietitian spelling

Good improvement of the IBC model, expanding its scope.

Having only people over the age of 36 among participants may be a bias, younger adults may bring different data. In the conclusions I suggest highlighting the need for research with young adults.

I also believe that there should be stages for assimilating the diagnosis and different understandings and behaviors experienced by the same person, it seems to me a process with different stages that vary individually over time.

I also found it surprising not to mention the change in body image (weight loss), which would be an excellent motivation, especially depending on the person's affective life, if they are married, divorced, etc. Failure to mention economic difficulties to carry out a healthy diet also reveals the profile of the consumption pattern of the participants and of the country in which the study was carried out..

It is clear from the results that despite being experienced personally, the change in diet is a social process, whether in a proximal or virtual mode. Lonely people seem to be less motivated which reveals that all human experience is necessarily collective, even when this is not evident.

The person's mood did not emerge from the data but it probably has significant importance, I suggest pointing out in the closing remarks. Other emotional difficulties such as depression and anxiety must also play a part.

Environmental aspects are very relevant, both in their physical and relational aspects (obesogenic environment and facilitators/saboteurs).

The article is very relevant to knowledge in the field in developed and social welfare countries. Even so, it brings important insights about the subject and that can motivate research in other countries. This is important to relativize the universality of theoretical models that deal with individual differences, there are also differences between nations and even within them.

Clear and very informative text that can positively influence patients but, above all, healthcare workers. As the article was too long, I suggest that the authors, in a final review, select and remove or condense information and clarification that are repeated

Reviewer #2: Dear Editor,

I believe this manuscript deserves publication. It adds to the existing literature on the Integrated Behaviour Change (IBC) model, as the authors put it: “to incorporate key constructs that underpin multiple processes known to guide behavioural decision making”.

The authors make an excellent use of COREQ guidelines and principles, which I believe enhances the reporting of qualitative research methods and findings. This is not frequent and I would like to congratulate the authors on that choice. It clearly facilitates the reviewer process and provides reassurance on the methods.

I would also like to emphasize the merit of authors accounting for their paper’s limitations, and I agree with them, including assumptions such as homogeneity in dietician performance and service quality.

Despite the above there are two issues that may need revision or reflection.

1. The authors claim (page 4, line 89) that “The exploration of such factors [individual difference factors and emotion determinants], alongside motivational, volitional, and implicit determinants of the IBC model, may pave the way for more elaborated integrated theories to be tested to advance knowledge on behavioural determinants and processes”.

This is a valid claim. I agree with the authors that the IBC model is not enough in explaining behaviours (and thereafter lifestyles[**]). This has occupied the agenda of sociologists for some time now (which is not being acknowledged by authors in their text). See for example the work by Cockerham and other authors on Life Choices and Life Chances, which studied the importance of agency and structure in shaping lifestyles.

The authors do acknowledge to some extent the importance of structural variables (beyond agency). This becomes very evident on page 27, line 537 [“One participant who worked away at sea had limited control over the food provided on site, therefore his agency to choose food for his diabetes was restricted beyond his control”]. I would argue this offers the authors the opportunity to mention the importance of socially defined dimensions (structure) as constrains to free will and choice. The authors provide other examples of such factors, and although they group them up as “Individual difference factors” they clearly refer to social factors (work, social class, income level, physical environment) limiting the array of options available there for individual choice. For example, being able to pay for a dietician or for good quality food is clearly influencing outcomes here.

In brief, although the first paragraph in the methods section seems to account for social factors at the time of understanding how reality is really constructed, this somehow vanishes as you progress in the text.

[** When discussing dietary behaviours among T2D patients (which is a chronic condition patients will live with for the rest of their lives) we should be aware that such behaviours are precursors of lifestyles, and lifestyles are less easy to modify, largely constructed through repetition and reinforcement.]

2. This second issue is closely related to the first. In S3 Table “Characteristics and demographics of participants in a qualitative study exploring the decision-making processes of dietary behaviour change after a diagnosis of type 2 diabetes”, the authors account for GENDER, SELF-REPORTED SOCIAL CLASS, and INCOME levels. I do not really see the need for that when such variables are mostly absent in the analysis. Do not take me wrong. I feel very strong about the importance of such variables (see point 1) and I would have favoured a more detailed analysis of data according to gender and social class, for example. I am just pointing out that these are two key dimensions that somehow have been neglected by the authors as explanatory variables.

6. PLOS authors have the option to publish the peer review history of their article (what does this mean?). If published, this will include your full peer review and any attached files.

Reviewer #1: **Yes: **Christiana Cabicieri Profice

Reviewer #2: **Yes: **Pedro Gallo

---

## [Author Response · Author response to Decision Letter 0]

27 Feb 2022

Thank you for the opportunity to revise and resubmit the manuscript, " Understanding dietary behaviour change after a diagnosis of diabetes: a qualitative investigation of adults with type 2 diabetes.", exclusively to PLOS ONE. 

We thank you and the reviewers for your consideration and suggestions that we feel have improved the manuscript. We therefore submit the revised version for your consideration. 

Sincerely,

Dr Roshan Rigby

On behalf of the author team.

---

## [Decision Letter · Decision Letter 1]

29 Nov 2022

Understanding dietary behaviour change after a diagnosis of diabetes: a qualitative investigation of adults with type 2 diabetes.

PONE-D-21-23420R1

Dear Dr. Rigby,

We’re pleased to inform you that your manuscript has been judged scientifically suitable for publication and will be formally accepted for publication once it meets all outstanding technical requirements.

Kind regards,

Jeffrey S. Hallam, Ph.D., FRSPH

Academic Editor

PLOS ONE

Reviewers' comments:

Reviewer's Responses to Questions

**Comments to the Author**

1. If the authors have adequately addressed your comments raised in a previous round of review and you feel that this manuscript is now acceptable for publication, you may indicate that here to bypass the “Comments to the Author” section, enter your conflict of interest statement in the “Confidential to Editor” section, and submit your "Accept" recommendation.

Reviewer #2: All comments have been addressed

2. Is the manuscript technically sound, and do the data support the conclusions?

Reviewer #2: Yes

3. Has the statistical analysis been performed appropriately and rigorously? 

Reviewer #2: N/A

4. Have the authors made all data underlying the findings in their manuscript fully available?

Reviewer #2: Yes

5. Is the manuscript presented in an intelligible fashion and written in standard English?

Reviewer #2: Yes

6. Review Comments to the Author

Reviewer #2: (No Response)

7. PLOS authors have the option to publish the peer review history of their article (what does this mean?). If published, this will include your full peer review and any attached files.

Reviewer #2: **Yes: **Pedro Gallo

---

## [Editor Report · Acceptance letter]

2 Dec 2022

PONE-D-21-23420R1 

Understanding dietary behaviour change after a diagnosis of diabetes: a qualitative investigation of adults with type 2 diabetes. 

Dear Dr. Rigby:

I'm pleased to inform you that your manuscript has been deemed suitable for publication in PLOS ONE. Congratulations! Your manuscript is now with our production department. 

Kind regards, 

on behalf of

Dr. Jeffrey S. Hallam 

Academic Editor

PLOS ONE